# Generation of the squamous epithelial roof of the 4[th] ventricle

**Florent Campo-Paysaa[1,2], Jonathan DW Clarke[1]\*, Richard JT Wingate[1]\***

[1]Department of Developmental Neurobiology, Institute of Psychiatry, Psychology and Neuroscience, King's College London, London, United Kingdom; [2]MRC Centre for Neurodevelopmental Disorders, Institute of Psychiatry, Psychology and Neuroscience, King's College London, London, United Kingdom

**Abstract** We use the transparency of zebrafish embryos to reveal the de novo generation of a simple squamous epithelium and identify the cellular architecture in the epithelial transition zone that ties this squamous epithelium to the columnar neuroepithelium within the embryo's brain. The simple squamous epithelium of the rhombencephalic roof plate is pioneered by distinct mesenchymal cells at the dorsal midline of the neural tube. Subsequently, a progenitor zone is established at the interface between columnar epithelium of the rhombic lip and the expanding squamous epithelium of the roof plate. Surprisingly, this interface consists of a single progenitor cell type that we have named the veil cell. Veil cells express *gdf6a* and constitute a lineage restricted stem zone that generates the squamous roof plate by direct transformation and asymmetrically fated divisions. Experimental restriction of roof plate expansion leads to extrusion of veil cell daughters and squamous cells, suggesting veil cell fate is regulated by the space available for roof plate growth.
DOI: https://doi.org/10.7554/eLife.38485.001

**\*For correspondence:**
jon.clarke@kcl.ac.uk (JDWC);
richard.wingate@kcl.ac.uk (RJTW)

**Competing interests:** The authors declare that no competing interests exist.

## Introduction

Epithelia are a fundamental but diverse animal tissue type. The origin and developmental programmes that generate the architecturally distinct types of epithelia (such as squamous, cuboidal and columnar) are poorly understood at both molecular and cellular levels. Furthermore, interfaces between architecturally distinct epithelia (known as epithelial transition zones) pose intriguing questions not only about their developmental origin, but also about how the growth dynamics of distinctly organised tissues can be integrated while maintaining the transitional architecture between them. While conspicuously under-explored, various lines of evidence point to epithelial transition zones as stem cell niches (*Mcnairn and Guasch, 2011*) and as important sites of cells of origin of cancer in the retina (*McKelvie et al., 2002*), lower gut (*Grodsky, 1961*) and cervix (*Fluhmann, 1960*).

Here we take advantage of the accessibility and transparency of the zebrafish embryo to reveal mechanisms underlying de novo generation of a simple squamous epithelium within the brain. While most of the neural tube is formed of a pseudostratified columnar neuroepithelium, the dorsal roof of the fourth ventricle in the rhombencephalon is composed of a simple squamous epithelium whose developmental programme is unknown. The boundary between the squamous flattened cells of the ventricular roof plate and the columnar cells of the neuroepithelium represent a most dramatic structural transition. The cellular architecture of this epithelial transition zone is not understood but could potentially be composed either of several cells with graded morphologies from columnar to squamous, or be composed of a single cell type whose morphology directly links a columnar cell to a squamous cell.

In the hindbrain, this epithelial transition is located adjacent to the rhombic lip, which is the progenitor pool for glutamatergic neurons of the cerebellum (*Machold and Fishell, 2005*; *Wang et al., 2005*) and hindbrain pre-cerebellar, auditory and vestibular systems (*Wang et al., 2005*). Rhombic lip progenitors are part of the columnar neuroepithelium and are specified by the bHLH transcription factor *atoh1* (*Ben-Arie et al., 1997*), which is induced (*Alder et al., 1999*; *Lee et al., 2000*) and maintained by Gdf secreted from the periphery of the adjacent roof plate (*Broom et al., 2012*). While atoh1-positive precursors at the rhombic lip give rise to populations of neurons, the adjacent *gdf*-expressing cells give rise to the simple squamous epithelium of the roof plate (*Currle et al., 2005*). This signaling and progenitor pool complex is maintained by cell-cell interactions (mediated by Delta-Notch signaling) at the interface between the roof plate and the neuroepithelium (*Broom et al., 2012*). While the molecular lineage and signaling required for formation of the squamous roof plate is known, the cellular organization of this epithelial transition zone, the precise identity and proliferative behaviours of roof plate progenitors at the margin of the roof plate, which act as a bi-directional organiser (*Broom et al., 2012*), remain unknown. These questions are crucial to understanding the origin of the squamous roof plate and how two architecturally distinct epithelia work together as a coherent sheet of cells to accommodate the changing shape of the hindbrain neuroepithelium and huge expansion of the enclosed fourth ventricle.

To answer these questions, we looked in detail at how cells behave in the dorsal midline of the embryonic hindbrain as it transforms into a squamous epithelium that tents over the fourth ventricle. To do this we used the zebrafish neural tube system, which allows high-resolution 4D confocal analysis of cellular development in the region of dorsal midline and the rhombic lip. We identify a novel population of cells at the dorsal midline of the neural tube that pioneer the de novo formation of the squamous roof plate and identify a novel roof plate stem cell with dynamic morphology. This cell forms a single cell bridge between the columnar rhombic lip and the squamous roof plate. This new cell type can transform into roof plate, generate new roof plate cells by asymmetrically fated cell divisions and undergo symmetric divisions to expand the stem cell pool. This population, which we name the 'veil' cell for its distinctive morphology, also expresses *gdf6a* and is thus a candidate for both the origin of roof plate development, and the cell type that confers organising signals to precursors the dorsal midline of the hindbrain neural tube. Experimental restriction of roof plate expansion leads to extrusion of squamous cells and veil cell daughters, suggesting that squamous cell number is regulated by the space available for roof plate growth.

## Results

### The ventricular roof plate emerges by an expansion of the dorsal midline

At 18hpf, prior to the emergence of the simple squamous epithelium of the roof plate, the hindbrain neural tube is a pseudostratified columnar epithelium that is indistinguishable from other regions of the central nervous system. The ventricle at this stage is a very narrow lumen at all dorsoventral levels as the left and right sides of the neural tube are initially very closely apposed to each other. However, over the next 12 hr the dorsal region of the ventricle expands greatly and the dorsal midline of the neural tube, specifically within the hindbrain, generates a new tissue type comprising a squamous epithelium that roofs over the expanding fourth ventricle. Both the squamous roof plate and columnar neuroepithelium can be visualized in live embryos of the Cdh2:(Cdh2-tFT) reporter line, and we followed the progress of ventricle expansion and roof plate formation from 18 hpf in a dorsal view (*Figure 1A*) using time-lapse confocal microscopy (*Figure 1—video 1*). Along with the ventricular cavity, the roof plate widens precociously at rhombomeres 2,4 and 6, while rhombomeres 3 and 5 lag a little behind, and gradually extends its caudal limit (the obex) to the level of somite 3/4. Transverse plane views show the squamous roof plate emerges at the dorsal midline as the columnar neuroepithelium folds outwards to enlarge the ventricular cavity (*Figure 1B*, *Figure 1—video 2*). The precise shape and squamous epithelial nature of the roof plate are revealed by focusing on the roof of the ventricle (*Figure 1C*). Transverse confocal sections show a rather abrupt transitional zone interface with the adjacent columnar neuroepithelium at the rhombic lip (*Figure 1B,D*), and this is confirmed by electron microscopy that suggests the interface may be composed of a single cell with a medial protrusion linking to the squamous cells (*Figure 1E*).

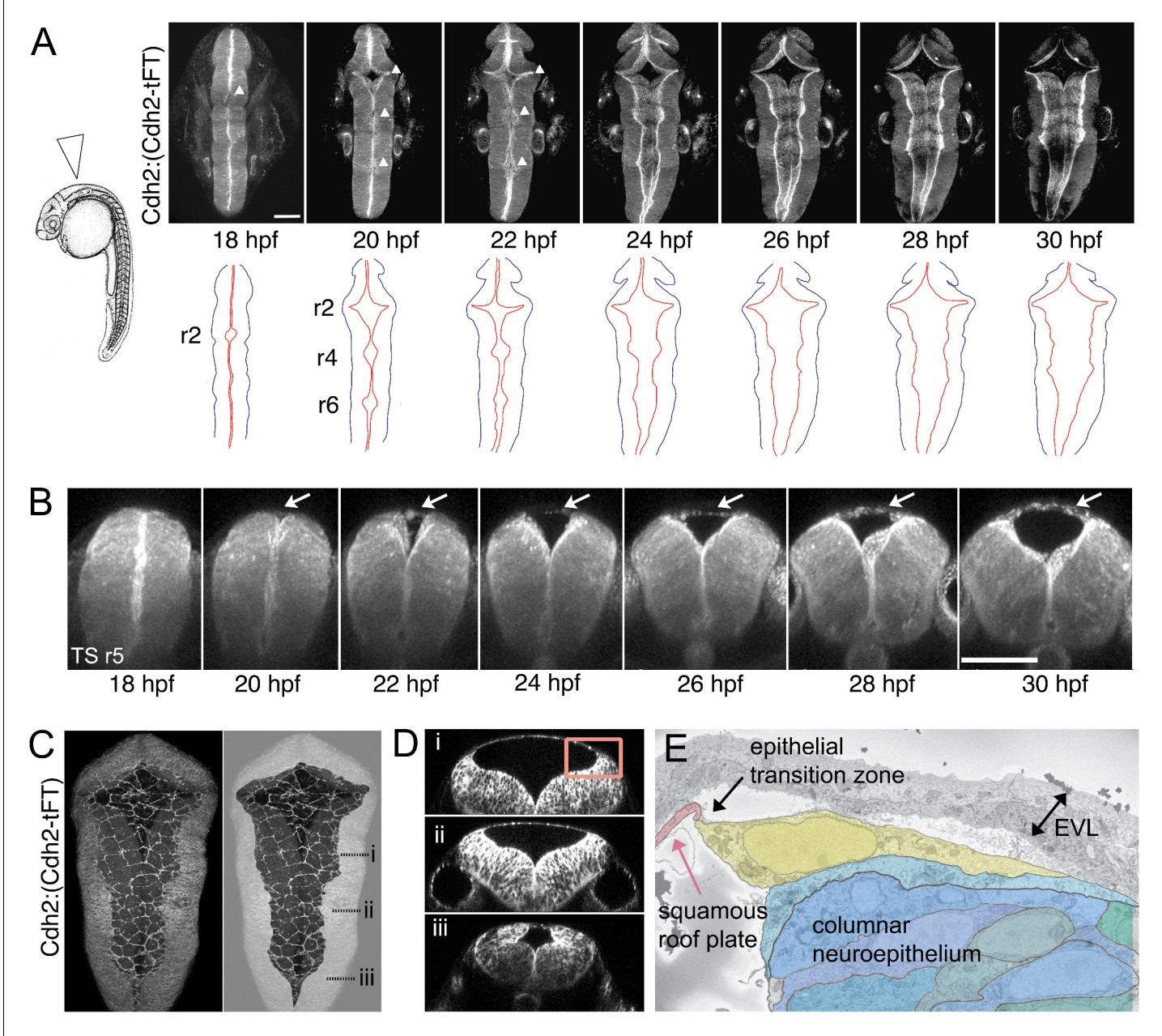

**Figure 1.** The squamous roof plate of the zebrafish hindbrain is generated from 18 hpf and links opposing rhombic lips. (**A**) Left. Diagram of 24 hpf zebrafish in lateral view. Arrowhead indicates location of hindbrain roof plate covering fourth ventricle and angle of imaging for time-lapse sequence. Right. Selected frames from time-lapse of Cdh2:(Cdh2-tFT) embryo (n = 15) tracking opening and expansion of fourth ventricle from 18 hpf to 32 hpf (see also *Figure 1—video 1*). White arrowheads on photomicrographs indicate the three points at which widening commences. Cdh2 is ubiquitous in neural cell membranes and most highly expressed at the ventricular surface. Brightest expression reveals the location of the rhombic lips which sit at interface between columnar neuroepithelium and the periphery of squamous roof plate. Outlines of the neural tube (black) and rhombic lip (red) are shown below. Roof plate widening is initiated in even-numbered rhombomeres (r), 2, 4, 6. (**B**) Outfolding of dorsal columnar epithelium and de novo appearance of the squamous roof plate (arrows) is revealed in transverse plane time-lapse of ventricle opening at level of rhombomere 5 (*Figure 1— video 2*). (**C**) Maximum projection of confocal sections of squamous roof plate and attached columnar epithelium in dorsal view of hindbrain in a 24 hpf Cdh2:(Cdh2-tFT) embryo. Columnar epithelium has been greyed out in right hand image to reveal shape of squamous roof plate. (**D**) Transverse plane reconstructions through levels indicated in (**C**) show shape of ventricle and domed squamous roof plate at different anteroposterior levels. (**E**) Electron micrograph of region indicated by inset in (**D**) of epithelial transition zone between squamous roof plate (pink) and columnar neuroepithelium of rhombic lip (blue and green). The most dorsal neuroepithelial cell, which abuts the roof plate at the transition zone between tissue architectures, is highlighted in yellow. Images in A-D are from Cdh2:(Cdh2-tFT) embryos. EVL, enveloping layer. Scale bars: 50 μm in A and B.

DOI: https://doi.org/10.7554/eLife.38485.002

*Figure 1 continued on next page*

*Figure 1 continued*

The following videos are available for figure 1:

**Figure 1—video 1.** Opening of the fourth ventricle Confocal time-lapse images show opening and inflation of the fourth ventricle in a Cdh2:Cdh2-tFT embryo, from 18 hpf to 32 hpf.

DOI: https://doi.org/10.7554/eLife.38485.003

**Figure 1—video 2.** Opening of the fourth ventricle and squamous roof plate formation.

DOI: https://doi.org/10.7554/eLife.38485.004

## Mesenchymal dorsal midline cells transform into pioneering squamous roof plate cells

Given the lack of an adjacent squamous epithelia to act as a source of cells, what is the origin of squamous cells of the ventricular roof plate? We imaged cells at the dorsal midline of the neural tube to discriminate between two potential mechanisms for the formation of this new squamous epithelium: a columnar to squamous transition of neuroepithelial cells versus a mesenchymal to epithelial transition of cells that lie just dorsal to the columnar neuroepithelium. At 17 hpf these two populations can be identified before the squamous roof plate forms. Dorsal columnar epithelial cells have aligned themselves on either side of the neural midline to generate the nascent ventricular surface (*Figure 2A* bottom panel and *Tawk et al., 2007*). Dorsal to this, a layer of overlapping irregular cells spans the midline, sandwiched between the columnar epithelium and the overlying enveloping layer (EVL) (*Figure 2A* top panel). In contrast to the polarised expression of the apical protein Pard3-GFP within the apical end feet of the columnar neuroepithelial cells (*Figure 2B* bottom), the irregular dorsal cells express little and patchy Pard3-GFP (*Figure 2B*).

We monitored the behaviour of both the columnar and more irregular cells at the dorsal midline during the initial period of ventricle widening between 18 and 26 hpf (regions mapped in *Figure 2C*). Over a period of 40 min, the irregular dorsal cells rearrange themselves so that there is no or very little overlap across the dorsal midline, thus forming a rectilinear interface at the dorsal midline (*Figure 2D,E*). Next, as the ventricle beneath them begins to expand the lateral membranes of some of these midline cells take on a very distinctive spiky appearance (*Figure 2E*). As the left and right sides of the underlying columnar epithelia move apart to enlarge the ventricle, the medial poles of these dorsal cells remain attached to each other and are 'left behind' to fill the potential space between columnar epithelia. As the ventricle enlarges over a period of 4 hr some of the dorsal cells gradually lose the spiky lateral protrusions that were contacting the basolateral domains of the columnar epithelia and the cells transform to a squamous, polygonal morphology thus forming the first distinctive squamous roof plate architecture (*Figure 2E*; *Figure 2—video 1*). In rare examples, irregularly shaped midline cells undergo pronounced rostrocaudal migrations before integrating into the expanding squamous roof plate (*Figure 2F*; *Figure 1—video 2*). Differentiation into the squamous roof plate is accompanied by maturation of their Pard3-GFP expression patterns: the irregular distribution of Pard3-GFP in cells prior to their transition into squamous roof plate is transformed into a continuous Pard3-GFP domain around the entire polygonal cell perimeter when they are properly part of the squamous roof plate (*Figure 2G*). In contrast to the irregular cells, the adjacent columnar epithelial cells were never observed to transform into squamous cells. Our time-lapse analysis hence shows that the squamous roof plate is pioneered by mesenchymal cells at the dorsal midline of the neural tube.

## A single, novel cell type links the squamous roof plate and columnar rhombic lip

The formation of the squamous roof plate also leads to a new interface between epithelial tissue types at the dorsal midline. The geometry of cells at this join presents a challenge to both tissues and the arrangement of cellular boundaries at this transition zone is unknown. How is a thin flat sheet of polygonal cells linked to thick columnar epithelium composed of spindle-shaped cells? To investigate whether the two epithelia are directly apposed at a specialized cell-cell interface, or whether the transition necessitates that formation of an intermediate cell type, we examined the morphology of cells at the interface between columnar neuroepithelium and squamous epithelial roof plate. To visualize cell morphology, we randomly labeled cells with the membrane tag Caax-

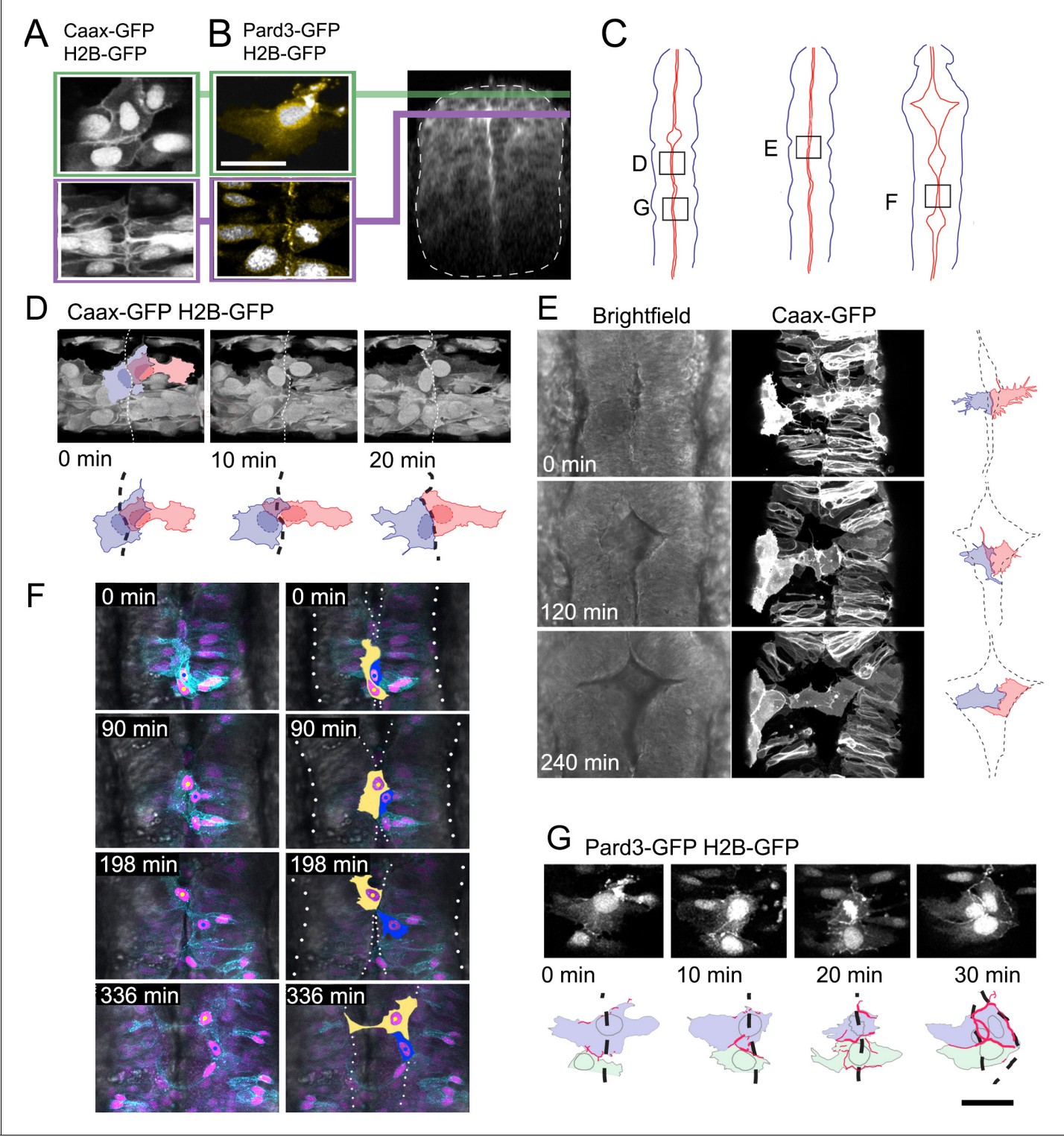

**Figure 2.** Dorsal mesenchymal cells pioneer the squamous roof plate. (**A**) Prior to ventricle expansion cells at dorsal midline of the neural tube are irregular and overlap the midline, while slightly deeper columnar cells align their apical end feet to the midline. (**B**) Pard3-GFP (yellow) is unpolarised in irregular dorsal cells but already polarized to the apical pole of columnar neuroepithelial cells that lie just deep to the dorsal surface of the neural tube. Transverse plane view of neural tube indicates level of sections in (**A**) and (**B, C**) Schematic diagrams of neural tubes show location of cells illustrated in panels (**D**) to (**G, D**) Three frames from time-lapse starting at 18hpf show two superficial cells at the dorsal midline (dashed line in drawings) initially overlap but later share a contiguous border that maps to the geometric midline just prior to opening of the ventricle. (**E**) Three frames from time-lapse

*Figure 2 continued on next page*

*Figure 2 continued*
show two spikey midline cells at dorsalmost aspect of neural tube transform directly into squamous roof plate cells as the ventricle opens over 4 hr. Dotted lines in drawings represent borders of roof plate (see *Figure 2—video 1*). (F) Four frames from time-lapse starting at 21hpf show irregular dorsal midline cell (nucleus marked by yellow dot on left hand images and cytoplasm filled yellow in adjacent right hand images) migrating along the dorsal tube midline to seed squamous roof plate as ventricle opens. A second roof plate pioneer is marked with blue dot and blue cytoplasm. Dotted lines in images show borders of roof plate and lateral edges of neural tube (see *Figure 2—video 2*). (G) Pard3-GFP protein is progressively allocated to cell-cell junctions surrounding a squamous roof plate cell during the opening of the ventricle. Scale bar in (G): 20 µm.
DOI: https://doi.org/10.7554/eLife.38485.005
The following videos are available for figure 2:
**Figure 2—video 1.** Pioneers of squamous roof plate cells in the hindbrain.
DOI: https://doi.org/10.7554/eLife.38485.006
**Figure 2—video 2.** Anterior migration of squamous roof plate pioneers.
DOI: https://doi.org/10.7554/eLife.38485.007

GFP and the nuclear tag H2B-RFP. This revealed cells at the columnar-squamous interface have a very distinctive morphology. These cells have a characteristic irregular membranous extension that drapes itself over the adjacent columnar rhombic lip cells and a cell body with nucleus that sits just medial to these columnar cells (i.e. just within the roof plate territory) (*Figure 3A,B*). The medial edge of these cells joins directly onto the squamous cells of the roof plate. These results show that the interface between epithelial types is populated by a single distinct cell type, which we term a 'veil cell' after its veil-like lateral membranous extension.

To define more precisely the relationship of veil cells to the columnar rhombic lip cells, we analysed veil cell morphology in the atoh1a:EGFP transgenic line that reveals rhombic lip cells. This shows that the veil cell's membranous extension is in intimate contact with the underlying *atoh1a*-positive cells of the rhombic lip (*Figure 3C*). Using Pard3-GFP expression to define the apical surface of veil cells, we find Pard3-GFP is excluded from the veil and forms a Pard3-GFP ring that outlines the apical end foot of the veil cells that connects both medially to the large interface with squamous cells of the roof plate and laterally to the mosaic of small apical end feet of the columnar epithelium of the rhombic lip (*Figure 3D*). The apical surface of each veil cell has a cilium that is close to their nucleus and protrudes into the ventricle (*Figure 3E*). Their cilium lies close to the organizing centre of their microtubule network. Within the veil, microtubules are distinctly polarized and organized into near linear arrays of microtubules (*Figure 3F*), in contrast to the radially organized microtubule cytoskeleton of cells lying fully within the squamous roof plate (*Figure 3G*).

## The veil cell population forms a dynamic, *gdf6a*-positive, single cell transition zone between squamous and columnar epithelia

To accommodate the expansion of the fourth ventricle and the squamous roof plate, we hypothesised that the cells at the interface of columnar and squamous epithelia may need to be quite dynamic. Therefore, to assess the behaviour of veil cells over time we used time-lapse microscopy. This revealed the lateral veil of these progenitors (that overlies the neuroepithelial rhombic lip cells) is highly motile and constantly remodels its borders. The mean length of the veil is 27 µm ± 2.8 (n = 20) and thus covers approximately 60% of the basolateral surface of the underlying columnar rhombic lip progenitors, which are 45 µm ± 3.2 long (n = 20). Despite the highly dynamic nature of individual veils, veil cells collectively form a continuous population at the interface of columnar and squamous tissues that maintains a consistent, if constantly adjusted contact with atoh1-positie cells of the rhombic lip (veils of adjacent cells are highlighted in *Figure 4A*; *Figure 4—video 1*).

Their location adjacent to the rhombic lip, raises the possibility that veil cells represent the organiser that induces (*Alder et al., 1999*; *Lee et al., 2000*) and maintains (*Broom et al., 2012*) atoh1 in adjacent rhombic lip. The molecular identity of this organiser as Gdf-expressing cells is long-established, however, the precise identity of these organiser cells (roof plate versus columnar epithelium) and their morphology has never been established. We find that veil cells not only express the growth factor *gdf6a* but, moreover, this expression distinguishes them from both the adjacent *atoh1*-positive rhombic lip cells and the majority of squamous roof plate cells (*Figure 4B*). In transverse section, *gdf6a* is seen to be strongly expressed in a single cell thick domain that precisely abuts the

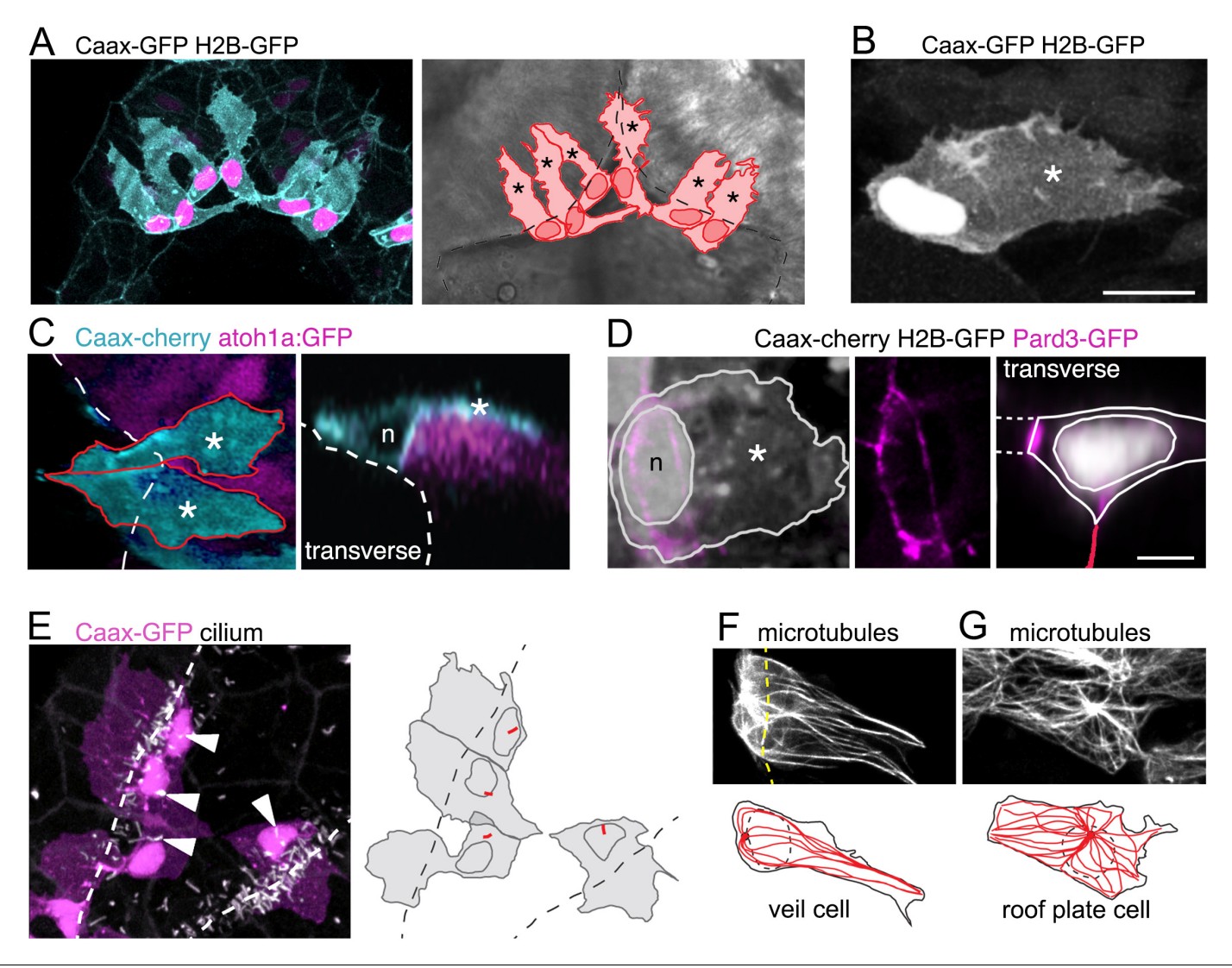

**Figure 3.** A cell with novel morphology, the 'veil' cell, occupies the transition zone between squamous roof plate and columnar neuroepithelium. (A) Dorsal view of six cells in the transition zone in the rostral pole of the hindbrain roof plate. Cells have distinctive morphologies with irregular lateral veils (asterisked in schematic to right) spanning the interface (dotted line) between roof plate and the columnar neuroepithelium. (B) Single veil cell with a nucleus placed at periphery of roof plate territory and a lateral veil (asterisk) extending over neuroepithelium. (C) Left. A dorsal view of two veil cells (cyan) with lateral veils (asterisks) extending over rhombic lip progenitors expressing atoh1a:GFP (magenta). Roof plate border shown with dashed line. Right. In transverse view reconstruction a veil cell presents a characteristically comma-shaped profile as its veil (asterisk) wraps itself over the basolateral surface of the atoh1-positive neural rhombic lip precursor. Apical surface of ventricle shown dashed, nucleus indicated with n. (D) Left: Dorsal view of veil cell, its nucleus (n) and veil (asterisk). Pard3-GFP expression shown in magenta. Middle: Single channel showing this cell's apical ring of Pard3-GFP. Right: Same cell reconstructed in transverse plane showing Pard3-GFP (magenta) expression at the ventricular interface with squamous epithelium (dashed lines) and apical surface of columnar epithelium (red line). (E) Cilia (white) of veil cells (magenta) are located close to nuclei and protrude into ventricle. Roof plate border shown dashed. Many cilia from columnar cells are also visible in the maximum projection. (F) Two adjacent veil cells expressing Dck1k-GFP show highly polarized microtubule networks, comprising a near linear array extending into the veils. (G) By comparison, central cell in this image is a squamous roof plate cell displaying radially organized microtubule array around a centrally placed focal point.

DOI: https://doi.org/10.7554/eLife.38485.008

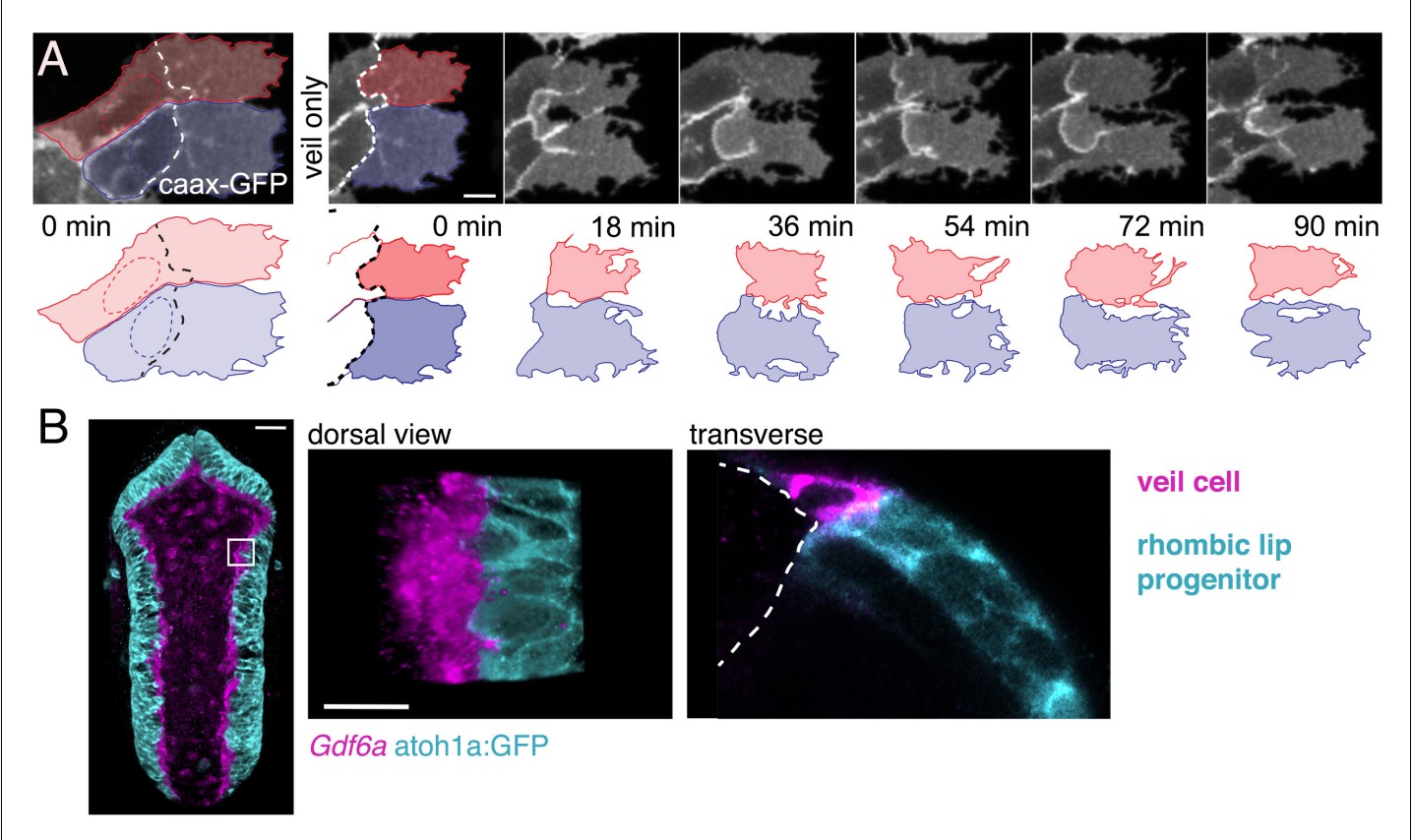

**Figure 4.** Veil cells comprise a morphologically dynamic, contiguous *gdf6a*-positive population that wraps over the rhombic lip. (A) Dorsal view of time-lapse sequence of two adjacent veil cells highlighting dynamics of their basolateral veils (blue and red, see also *Figure 4—video 1*). Outline drawings of veil dynamics shown below. (B) Left. Dorsal view of hindbrain showing in situ hybridisation for *Gdf6a* (magenta) and atoh1a-GFP rhombic lip progenitors (cyan). Right. High magnification of inset area in (g) and a transverse plane reconstruction.

DOI: https://doi.org/10.7554/eLife.38485.009

The following video is available for figure 4:
**Figure 4—video 1.** Veil dynamic during interphase.

DOI: https://doi.org/10.7554/eLife.38485.010

expression domain of *atoh1* in the columnar rhombic lip (*Figure 4B*). This suggests that veil cells comprise a single cell wide signalling centre that organises the adjacent rhombic lip.

## Some veil cells transform directly into squamous roof plate cells

Previous work in amniotes has shown that the Gdf expressing cells not only comprise the roof plate organiser but also that Gdf-expressing cells give rise to the overwhelming majority of roof plate tissue (*Currle et al., 2005*). This would imply that the single line of *gdf6a* expressing veil cells gives rise to the entire roof plate in zebrafish. To determine whether this is the case, we followed veil cells using time-lapse confocal microscopy for up to 12 hr from 22 hpf (n = 124 embryos, 1100 hr total imaging). We find some veil cells can directly transform into squamous roof plate cells in a similar manner to the pioneering dorsal cells that initiate roof plate expansion. Veil cells are able detach gradually from the columnar rhombic lip by a slow, progressive retraction of their veil (**2 to 3** hr) and intercalate directly into the roof plate (n = 33, *Figure 5A*; *Figure 5—video 1*). The cytoplasm of the veil appears to be reallocated medially to form the squamous morphology. During this process, the nucleus initially remains static as the rest of the cell reorganises around it. The direct transformation of veil cells into squamous roof plate cells occurs predominantly in the lower rhombic lip (*Figure 5B*) and, by implication from in situ labelling, is accompanied by the downregulation of *Gdf6a*. Importantly, we did not see any squamous roof plate cells transform back into veil cells, nor neuroepithelial

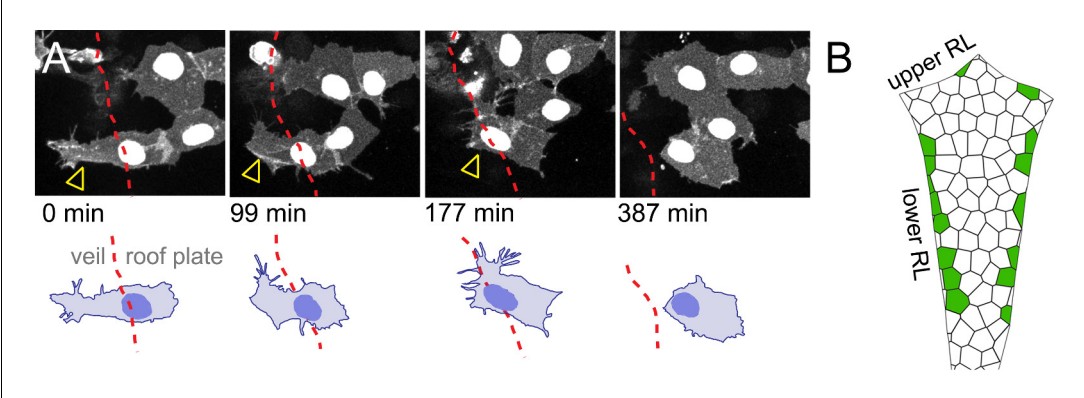

**Figure 5.** Veil cells can transform into definitive squamous roof plate cells. (**A**) Four frames from a time-lapse of caax-RFP labeled veil cell starting at 22 hpf transforming directly to a roof plate cell. Retraction of the veil from columnar epithelium is indicated with arrowheads. Dashed red line is boundary between columnar epithelium to its left and squamous epithelium to its right (see *Figure 5—video 1*). (**B**) Schematic shows map of 16 veil cells that transformed directly to roof plate cells, most lie in the lower rhombic lip.

DOI: https://doi.org/10.7554/eLife.38485.011

The following video is available for figure 5:

**Figure 5—video 1.** Direct transformation of a veil cell into a squamous roof plate cell, Related to *Figure 5* .

DOI: https://doi.org/10.7554/eLife.38485.012

cells transforming into veil cells in over 1000 hr of recorded development. This suggests that if the veil cell population remains stable it must be replenished exclusively by division within the veil cell population.

## Veil cells can also generate squamous roof plate cells via asymmetric divisions and self-renew via symmetric divisions

We next set out to confirm whether veil cell divisions replenish the population of veil cells and also whether veil cell divisions might also directly give rise to new roof plate cells. To assess whether either or both these possible modes of proliferation occur, we monitored the veil cell population for both frequency and mode of cell divisions from 20 hpf onwards. This data revealed many divisions of veil cells at the periphery of the squamous epithelium (*Figure 6A*). Analyses of the daughters of these divisions revealed veil cells are capable of both self-renewing asymmetric divisions and symmetrically fated divisions. By contrast, we only observed 11 cell divisions within the squamous epithelium itself over the course of 124 time-lapse recordings (approximately 1100 hr recorded at 24–36 hpf), suggesting that such mitoses are extremely rare.

Time-lapse analysis shows that veil cell mitosis is preceded by a remodelling of the cell's lateral veil, which undergoes a cytoplasmic retraction leaving spikey, skeletal adhesions at the periphery of the veil (n = 27, *Figure 6B*; *Figure 6—video 1*). However, unlike their neighbouring columnar neuro-epithelial cells, there is no interkinetic nuclear migration accompnying the veil cell cycle. In asymmetrically fated divisions (*Figure 6—video 2*, n = 26), while one daughter differentiates into a squamous roof plate cell, the more lateral daughter remains in the transition zone to replenish the veil cell population (*Figure 6B*). These asymmetrically fated divisions are predominantly distributed along the lower rhombic lip (*Figure 6C*).

We also found a smaller proportion of veil cells that divide symmetrically to generate two veils cells (n = 21, *Figure 6D*, *Figure 6—video 2*). During symmetric divisions, one daughter cell generally inherits the veil, while the other daughter generates a new lateral veil extension after mitosis. Symmetrically fated cell divisions are more frequently found along the upper rhombic lip (*Figure 6E*). Veil cell divisions never produce daughters that populate the columnar epithelium of the rhombic lip, thus through symmetric and asymmetric divisions they behave as a unipotent stem zone for the squamous roof plate.

An analysis of location of divisions suggest the mode of veil cell division may be related to the differential axes of growth at different locations along the hindbrain. Most asymmetric divisions occur

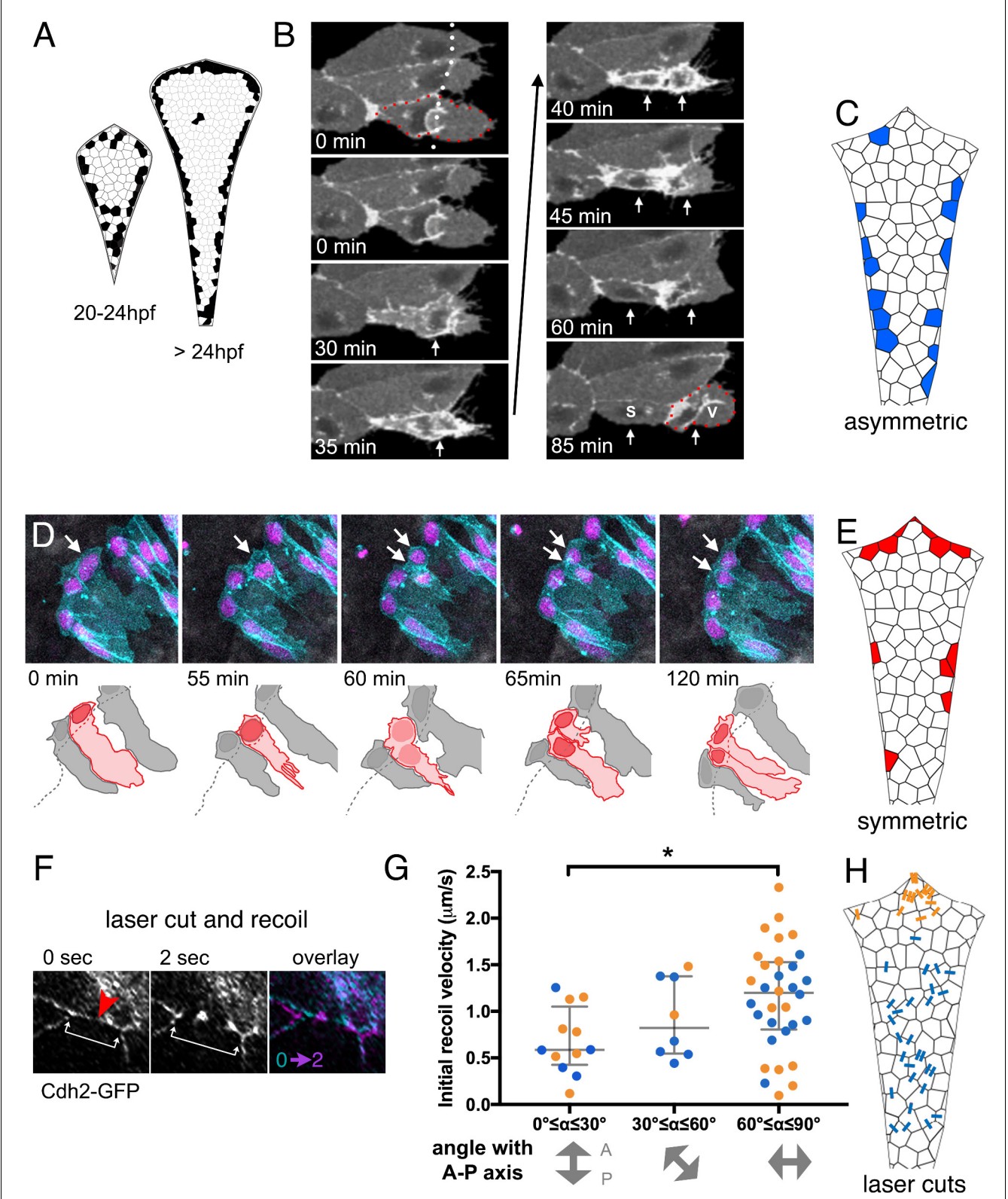

**Figure 6.** Veil cells can undergo self-renewing asymmetric divisions and symmetric proliferative divisions. (**A**) Schematics of roof plate that cumulatively map the distribution of cell divisions between 20 and 24 hpf and from 24 to 36 hpf from 125 embryos. (**B**) Time-lapse sequence of veil cell (outlined by red dots) during an asymmetric division (see also *Figure 6—video 1*) with cleavage plane parallel to roof plate border (dotted white line). Two timepoints in mitosis indicated by single white arrow at 30 and 35 min. Position of two daughters indicated by white arrows from 40 min onwards.
*Figure 6 continued on next page*

*Figure 6 continued*
Medial daughter becomes a squamous roof plate cell (marked S) and lateral daughter becomes a veil cell (marked V and outlined by red dots). Images also contain a second veil cell and an established squamous cell that have been indicated with reduced brightness in the drawing. (C) Schematic map of location of 26 asymmetrically fated veil cell divisions. (D) Five frames from time-lapse of a symmetrically fated veil cell division starting at 23 hpf that generates daughters with same fates (two veil cells). Arrows indicate mother and daughter veil cells. One daughter inherits the original basolateral veil while the other rapidly regenerates a veil after mitosis (see also *Figure 6—video 2*). (E) Schematic map of 19 symmetrically fated veil cell divisions. (F) Time-lapse sequence (see also *Figure 6—video 3*) showing laser cut of a cell-cell junction in the roof plate of a Cdh2-GFP embryo at 24hpf. Site of laser ablation is indicted with a red arrowhead. Arrows indicate positions and recoil of adjacent tricellular vertices. (G) Quantification of initial recoil velocity after laser cut as a function of the angle to the anteroposterior axis shows a significantly higher velocity following parallel versus perpendicular cuts (Two-tailed Mann-Whitney, p=0.0113). Recoil in the region proximal to the upper rhombic lip (orange) is similar to that adjacent to the lower rhombic lip (blue). (H) Map of positions and angles of laser cuts.
DOI: https://doi.org/10.7554/eLife.38485.013
The following videos are available for figure 6:
**Figure 6—video 1.** Asymmetric division of veil cell.
DOI: https://doi.org/10.7554/eLife.38485.014
**Figure 6—video 2.** Symmetric division of a veil cell.
DOI: https://doi.org/10.7554/eLife.38485.015
**Figure 6—video 3.** Laser cutting of cell-cell interface in the squamous roof plate, Related to *Figure 6*.
DOI: https://doi.org/10.7554/eLife.38485.016

in the lower rhombic lip (*Figure 6C*): Their daughters become aligned along the mediolateral axis and we predict this is the dominant axis of tension at the lower rhombic lip as the left and right sides of the neuroepithelium fold outwards to expand the ventricle (*Figure 1A,B*). Most symmetric divisions occur in the upper rhombic lip (*Figure 6E*) and their daughters become aligned along the rhombic lip rather than across it. We predict that this also reflects the dominant axis of tension at the upper rhombic lip as this border of the roof plate expands mediolaterally during ventricle expansion (*Figure 1A*). The separation of symmetric daughters suggests they are potentially being pulled apart along the border of the roof plate, while the separation of asymmetric daughters suggest they are being pulled apart across the border of the roof plate.

If 'pull' does play a role in daughter cells fate, we would expect the pattern of mechanical tension across the developing roof plate to be higher along the mediolateral axis. To determine whether this is the case we performed multiphoton laser cuts at cell-cell interfaces in the roof plate (*Figure 6F*; *Figure 6—video 3*) and measured initial recoil velocities (*Figure 6G*) of membranes along the various orientations across the squamous roof plate (*Figure 6H*). This mapping revealed significantly higher recoil speeds across the mediolateral axis of the roof plate compared to recoil speeds along other axes of the epithelium.

These results firstly show that veil cells can directly contribute to roof plate growth by division while also replenishing the veil cell population as needed for hindbrain growth. Secondly, the location of veil cell transformation (*Figure 5*) and the distribution of asymmetric and symmetric divisions (*Figure 6C,E*) all correlate with the local predominant axis of mechanical tension in the expanding roof plate. This suggests that mode of division may be a response to the mechanical demands of morphogenesis and not a product of regional patterning per se. We have not been able to follow a veil cell through multiple rounds of division, so we do not yet know whether a single veil cell can undergo both asymmetric and symmetric division.

## Restriction of roof plate growth leads to veil cell daughter extrusion

In a small number of cases (n = 3) of symmetrically fated veil cell divisions, the daughter cell that generates a new veil positions its veil directly over the veil of its sister cell. In these cases, the second veil fails to establish contact with the rhombic lip and this daughter cell undergoes extrusion from the epithelium and subsequent cell death (*Figure 7A*). This raises the question of whether cell numbers in the roof plate are regulated by elimination in response to limiting space within the veil cell domain or perhaps in the squamous roof plate itself. To address this, we experimentally created a situation where wild type veil cells border a region of roof plate whose growth has been restricted. To do this, we turned to an experimental model that inhibits ventricle opening in rhombomeres 3 and 5 and consequently restricts the growth of the squamous roof plate in adjacent rhombomeres (especially rhombomere 4). Krox20-driven expression of a dominant negative version of Rab11a

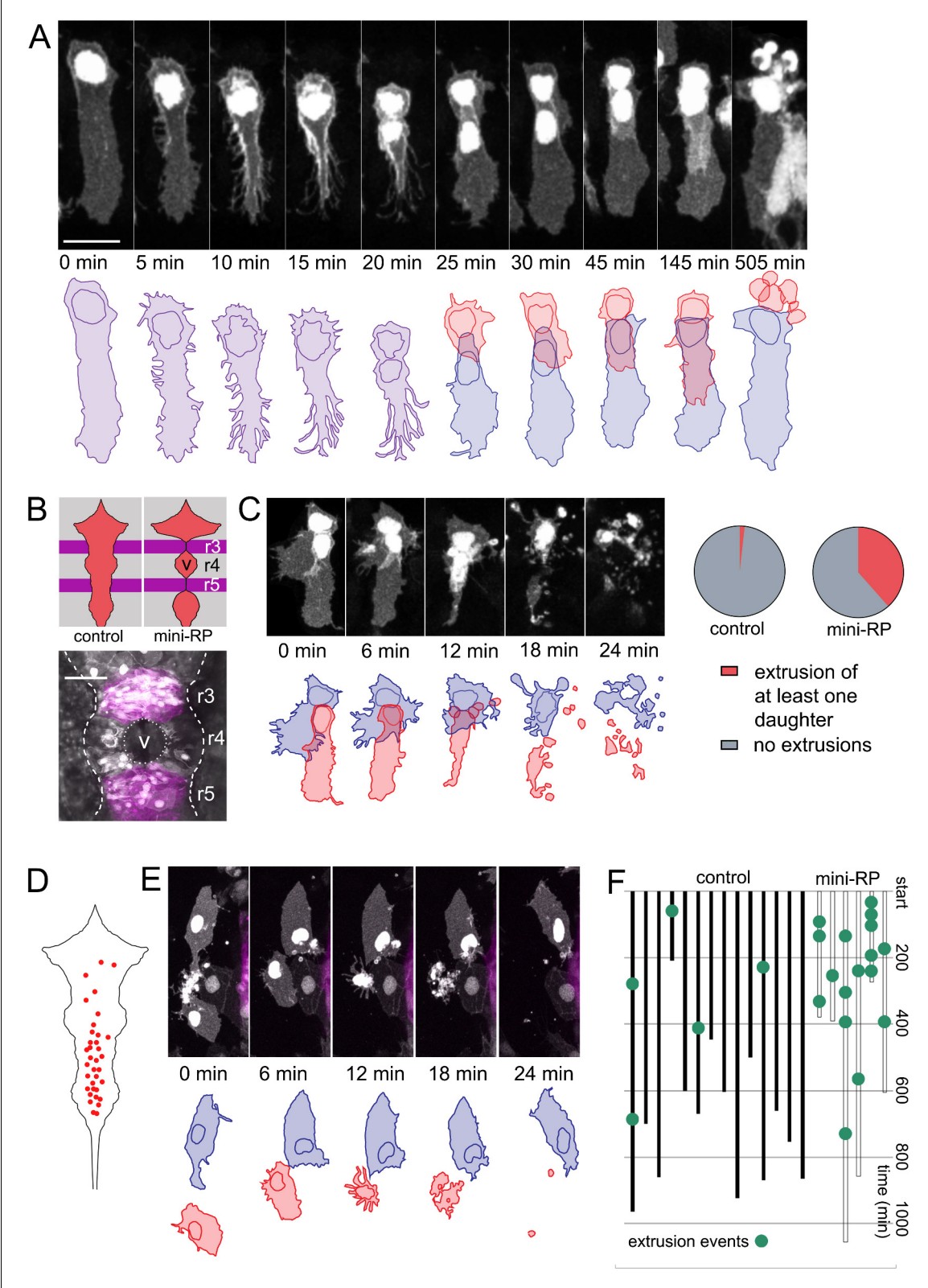

**Figure 7.** Space-regulated cell extrusions in the roof plate. (**A**) Time-lapse frames of veil cell division starting at 24 hpf with cleavage plane close to parallel to roof plate border generates daughters with same fates (two veil cells). The lateral daughter (blue in schematic) inherits the basolateral veil while the medial daughter (red in schematic) attempts to regenerate a new veil but becomes extruded. A schematic representation of the sequence is shown in the bottom panel. (**B**) Schematic of the dorsal hindbrain in control and mini-RP (Krox20 x Rab11dn) embryos, showing the failure in ventricle

*Figure 7 continued*

opening at the level of rhombomeres 3 and 5 and consequent restriction of ventricle and roof plate in rhombomere four in the latter. Photomicrograph is maximum projection of rhombomeres 3, 4 and 5 in a mini-RP embryo showing mini-ventricle in rhombomere four and lack of ventricles in rhombomeres 3 and 5 (magenta). (C) Time-lapse sequence of the extrusions of two sister veil cells in the roof plate of rhombomere four in a mini-RP embryo. A schematic of the sequence is shown below. Pie charts show the quantification of the proportion of veil cell divisions producing extruding daughter cells, in control (n = 21) and mini-RP embryos (n = 13). (D) Schematic shows map of 35 extrusions of squamous roof plate cells in control embryos. (E) Time-lapse sequence of a squamous cell (red cell in schematic) being extruded from roof plate of rhombomere 2, in a mini-RP embryo. Top cell in images (and blue in schematic) is a veil cell lying over rhombic lip. (F) Temporal maps of squamous cell extrusions in the roof plate of rhombomere 4, in control (black lines) and mini-RP (white lines) embryos. Each vertical line represents a log of cell extrusion events during a single time-lapse recording. Scale bars: 10 µm in A.

DOI: https://doi.org/10.7554/eLife.38485.017

(Rabl11a-S25N) in rhombomeres 3 and 5 inhibits ventricle formation in these segments (*Buckley et al., 2013*) and non-autonomously restricts expansion of the ventricles in adjacent segments, particularly rhombomere 4 (*Figure 7B*). Importantly both the veil cells and squamous roof plate of rhombomere four are wild-type, so their behaviour should only be modified by the physical constraints imposed by the neighbouring segments.

Following this manipulation, zebrafish transiently display a hindbrain with three small squamous roof plates, rather than one large diamond shape epithelium. We then quantified the behaviour of veil cells that generate the small roof plate at the level of rhombomere four and found that asymmetrically fated divisions at the periphery of this mini roof plate are frequently followed by one or both daughter cell extrusions (*Figure 7C*). We quantified extrusion as a proportion of overall asymmetric divisions and found that 38% of asymmetrically fated divisions are followed by extrusion in the mini roof plates (n = 5/13 divisions) versus only 5% in wild-type roof plates (n = 1/21 divisions) (*Figure 7C*). This suggests physical restriction of roof plate expansion may not limit the number of veil cell mitoses but does control the fate of their daughters in a manner that suggests veil cell fate is regulated by the space available for roof plate growth.

In addition to the newly generated daughters of veil cell divisions, time-lapse analyses reveal definitive squamous epithelial cells are also sometimes extruded from the squamous epithelium of wild-type roof plates. In fact, we found that extrusions preferentially occur in the posterior region of the squamous roof plate. This posterior region is characteristically narrower than the more anterior region of the roof plate (*Figure 7D*). Each extrusion characteristically takes between 15–20 min (*Figure 7E*) and were infrequent events within a given time-lapse recording (*Figure 7F*, left). We used this data as a baseline to compare to the rate of squamous roof plate cell extrusion in the in the mini rhombomere four roof plate of the Krox20-Rab11dn embryo. When instances of cell extrusion are logged in time-lapse recordings of mini roof plates (*Figure 7F*, right), squamous cells are 10 times more frequently extruded (a total of 17 extrusions in 3560 min of imaging, that is 1 extrusion every 200 min) than in rhombomere four territory in controls (5 extrusions in 9635 min or 0.1 extrusions every 200 min). Taken together these data suggest the survival of veil cell daughters depends on available space both within the veil cell domain itself and within the squamous epithelium. Thus, extrusion may represent an adaptive mechanism that is able to regionally regulate the number of squamous roof plate cells according to the area of the roof plate itself.

## Discussion

Ventricle formation is a correlate of brain morphogenesis in all vertebrates. In the hindbrain region this involves the generation of a thin squamous roof over the expanding fourth ventricle. We have used the hindbrain in zebrafish as a model for this process and defined a new progenitor population and characterized its behaviour as it generates the expanding squamous roof plate. The morphology of these progenitors, which we call veil cells, is adapted to bridge the interface between columnar and squamous epithelia. Veil cells are the architectural link between the morphologically very different columnar and squamous epithelia. In addition to linking these epithelia together they also generate the cells of the simple squamous epithelium that roofs over the expanding ventricle and organise gene expression in the neighbouring rhombic lip. Our results suggest that the morphology and proliferative capacity of veil cells make them ideally adapted to broker the dramatic

morphogenetic transformation of an initially slit-like lumen at the centre of the neural tube into the expansive diamond shaped fourth ventricle.

To our knowledge this is the first characterization of the de novo generation of a simple squamous epithelium in any in vivo system. In addition to forming a stem zone for the squamous roof plate, veil cells have a specialized, hybrid and dynamic morphology that allows them to form a single cell link between the columnar epithelial organization of the rhombic lip and the simple squamous sheet of the roof plate. Stable addition and integration of veil cell daughters into the squamous epithelium is regulated by cell extrusion and correlates with the space available, while the mode of veil cell division correlates with the mechanical properties of the epithelium.

## Origin of squamous pioneers

Our data show the cells that pioneer the squamous roof plate derive from a mesenchymal population that lie immediately dorsal to the midline of the neural tube. These cells are not part of the epithelial neural tube; they have a spiky mesenchymal morphology, and some are able to migrate along the dorsal midline before epithelialising and integrating into the neural tube. By a mechanism we do not understand, these pioneers undergo a mesenchymal to epithelial transition to fill the potential space left behind as the left and right rhombic lips move apart to generate the expanding ventricular space. As they fill this space they establish non-overlapping boundaries and begin to build the characteristic simple squamous morphology of the thin epithelial roof plate. The early mesenchymal nature and position of these pioneers above the dorsal midline of the neural tube suggest they share some characteristics with neural crest cells.

## Veil cells

Once the pioneer cells have colonized the territory of the nascent squamous roof plate this tissue grows by addition of squamous cells from progenitor cells at the periphery of the territory. We call these progenitors veil cells after their characteristic morphology. Veil cells form a self-renewing, unipotent stem zone for the squamous roof plate that expresses the secreted factor Gdf6a. We do not know whether veil cells are a heterogeneous population, some of which divide only symmetrically and others only asymmetrically, or whether a veil cells is capable of switching proliferation mode as required by growth. Importantly, we have never seen a columnar rhombic lip progenitor transform into a veil cell or *vice versa*, nor do squamous roof plate cells ever revert to a veil cell phenotype. The majority of the ventricular roof plate in mammals derived from a Gdf lineage (*Currle et al., 2005*) and our work in the fish gives, for the first time, a cellular and behavioural identity to these cells.

Unlike other simple squamous epithelia, such as the EVL cells in the zebrafish embryo (*Campinho et al., 2013*), which grow by mitoses within the sheet of squamous cells itself, the roof plate grows almost exclusively by addition of cells from its peripheral veil cell progenitor zone. On the assumption of a conservation of function across species, *Gdf6a*-positive veil cells may also act as a classic boundary organizer that regulates growth of both neighbouring columnar neuroepithelial cells and the squamous roof plate (*Broom et al., 2012*). The peripheral location of the veil cells around the squamous roof is thus perfectly located to integrate growth of these two functionally and architecturally distinct but physically coupled neural territories. The dependence of squamous roof plate on a Gdf6a lineage is reminiscent of the requirement for other TGFβ superfamily members in the development of squamous cells in the epithelium of the Drosophila egg chamber (*Brigaud et al., 2015*; *Duhart et al., 2017*).

## Space restrictions induce cell extrusion to regulate squamous cell numbers

The area of squamous roof plate is precisely confined by the adjacent columnar neuroepithelium, and the available space within this boundary expands during development as the neuroepithelium on left and right sides move apart to expand the volume of the fourth ventricle. Coordinating the growth of the squamous epithelium with the outfolding of the columnar neuroepithelium is achieved by regulating squamous cell numbers via a mechanism that recognizes the space available within the squamous roof plate. Regulating the number of squamous cells could be achieved by regulating the number of veil cell mitoses or by regulating how many of the daughters of veil cell divisions are

stably incorporated into the epithelium. Our data shows that extrusion of cells is one mechanism that regulates growth of the squamous epithelium. Extrusion operates at two levels in this system: One is to eliminate daughter cells soon after veil cell mitosis before they can properly integrate into the epithelium and the other, similar to epithelial extrusions in other systems (*Gu and Rosenblatt, 2012*; *Kocgozlu et al., 2016*; *Marinari et al., 2012*) is extrusion from within the squamous epithelium itself. These extrusion mechanisms suggest the rate of veil cell mitoses may be hard-wired and unresponsive to local growth needs, a hypothesis it will be important to test in the future.

## Tissue tension correlates with symmetry of veil cell divisions

Although our data suggest the overall rate of veil cell divisions may not be plastic, the local differences in proportions of asymmetric and symmetric divisions in the upper and lower rhombic lip suggest this choice may be related to local tissue architecture. Symmetric divisions are more prevalent along the upper rhombic lip, which is oriented mediolaterally and potentially stretched along this axis as the roof plate grows. Asymmetric divisions are more prevalent along the lower rhombic lip, which is oriented longitudinally and more likely to be under tension orthogonal to its long axis. The orientation of local tensile forces could potentially regulate spindle orientation during veil cell mitoses (*di Pietro et al., 2016*; *Fink et al., 2011*; *Nestor-Bergmann et al., 2014*) and this could regulate symmetry or asymmetry of daughter cell fate.

Whether the modes of division are regulated by tissue tension or not, the distribution of division types is likely to contribute to tissue morphogenesis. Symmetric divisions generate two veil cells that lie adjacent to each other at the roof plate border and thus potentially increase its length. This is consistent with the mediolateral expansion of the upper rhombic lip between 20 and 30 hpf (*Figure 1A*). Asymmetric divisions do not increase the number of veil cells and therefore do not affect the length of the roof plate border in the lower rhombic lip (*Figure 1A*), but they allow the left and right borders at the lower rhombic lips to move apart by generating new squamous cells to increase the width of the roof plate.

## Veil cells form a single cell epithelial transition zone

The interface between the simple squamous cells of the ventricular roof plate and the columnar cells of the rhombic lip represent a dramatic structural transition. Intriguingly the unique morphology of veil cells allows them to form a single cell bridge between the columnar and simple squamous epithelia. Thus, in addition to forming the unipotent stem zone of the squamous roof plate, veil cells provide the cellular architecture required to knit morphologically distinct epithelia together in this epithelial transition zone. In this context the term 'transition zone' does not imply that veil cells are a transitory morphological state as cells transit from columnar to squamous morphologies (our results show columnar neuroepithelial cells simply do not undergo that transition), rather the 'transition zone' is a structural link between diverse epithelia.

Epithelial transition zones are found in several tissues of the body and have attracted attention both as potential stem cell niches and because of their association with tumour formation in the gut and cervix. Epithelial transition zones are undeniably important but poorly studied regions of the body (*Mcnairn and Guasch, 2011*) and our work identifies the developmental origin of an epithelial transition zone for the first time and confirms this as a stem zone for a squamous epithelium. We speculate the unique architecture and dynamic behaviour of veil cells might be envisaged as a potential 'risk' in development. Failure to contain over-proliferation through extrusion or effectively translate stretch into coordinated cell division might lead to deregulated cell division. Accordingly, 17% of medulloblastomas (the predominant cancer of hindbrain and cerebellum) can be attributed to cells of origin at the rhombic lip epithelial transition zone (*Gibson et al., 2010*; *Kool et al., 2012*): almost as significant as the 25% of cancers that arise in granule cell derivatives of the upper rhombic lip. For the latter, unregulated proliferation can be attributed to a disruption of transit amplification regulation (*Goodrich et al., 1997*; *Wallace and Raff, 1999*; *Wechsler-Reya and Scott, 1999*). For the former, tumour formation may be triggered by a failure to respond correctly to mechanical cues for proliferation control.

In summary, our work introduces a new vertebrate model of epithelial morphogenesis and uncovers a novel neural progenitor – the veil cell. We show for the first time, the de novo generation of a squamous epithelium in vivo, the developmental origin of an epithelial transition zone cell type

and characterize its modes of cell division. We show that its dynamic properties are adapted to integrate morphogenesis of the squamous roof plate with the growth and morphogenesis of its surrounding columnar neuroepithelium.

## Materials and methods

### Fish lines

The Tg(atoh1a:EGFP) (*Kani et al., 2010*) and Cdh2:(Cdh2-tFT) (*Revenu et al., 2014*) lines were kindly donated by Masahiko Hibi and Darren Gilmour respectively. To prevent ventricle opening in rhombomeres 3 and 5, we crossed a UAS-inducible dominant-negative Rab11a line (Tg(UAS: mCherry-Rab11a S25N)mw35) with a line in which the optimised Gal4-activator, KalTA4, is driven by Krox20 specifically in rhombomeres 3 and 5 (tg(Krox20–RFP–KalTA4). This resulted in expression of Rab11a–S25N specifically in rhombomeres 3 and 5 as described previously (*Buckley et al., 2013*).

### Embryo care

Embryos were collected, staged and cultured according to standard protocols (*Kimmel et al., 1995*). All procedures were carried out with Home Office approval and were subject to local Ethical Committee review.

Expression constructs pCS2 +vectors containing cDNA for the following genes were linearised and mRNA synthesised with Ambion mMessage mMachine System from the sp6 promoter (AM1340): partitioning defective 3-GFP and RFP from zebrafish (pard3-GFP and pard3-RFP), histone H2B-GFP and RFP from human (H2B-GFP and H2B-RFP), GFP and CHERRY-CAAX from human (GFP-CAAX and CH-CAAX), end-binding protein 3-GFP from human (EB3-GFP), doublecortin-like kinase-GFP from human (Dck1k-GFP).

### Embryo injections

Embryos were injected using standard injection protocols. For ubiquitous distribution of mRNA, embryos were injected at a 1 to 4 cell stage. In order to image individual cells, embryos were mosaically labelled by injecting mRNA into 1 blastomere of a 16 to 64 cell stage embryo. pard3 construct mRNA was injected at 50 to 100 pg per embryo. mGFP, mRFP, CH-CAAX and H2B-RFP were injected at 50-100pgs per embryo. EB3-GFP and Dck1k-GFP were injected at 10–30 pg per embryo.

### Immunohistochemistry and in situ hybridisation

For in situ hybridisation and immunostaining embryos were fixed with 4% paraformaldehyde for 2 hr at room temperature. Embryos hybridised with an RNA probe for Gdf6a (V.Prince, University of Chicago) using standard protocols. Chicken anti-GFP (Sigma: 1:1000) was used for immunostaining in the atoh1a:gfp line with the appropriate alexa conjugated secondary antibody (Sigma: 1:500).

### Time-lapse confocal imaging and processing

Embryos were mounted in low melting point agarose and confocal time-lapse movies were made at 28.5°C as previously described using a Leica SP5 confocal and water dipping x25 and x40 objectives (*Tawk et al., 2007*). Data was collected from the hindbrain regions. Some adjacent cells have been cropped from the images to increase clarity of the cells of interest. Images were processed using Volocity and ImageJ.

### Electron microscopy

Embryos were fixed in 2.5% glutaraldehyde, 2% paraformaldehyde, 2 mM CaCl2 in 0.1 M cacodylate buffer (pH 7.4) overnight at 4°C. Hindbrains were dissected out and fixed in 1.5% potassium ferrocyanide:2% osmium tetroxide in cacodylate buffer for 1 hr at room temperature 1.5% potassium ferrocyanide:2% osmium tetroxide in cacodylate buffer for 1 hr at room temperature. Tissue was then thoroughly rinsed in distilled water and incubated in 1% aqueous thiocarbohydrazide for 4 min. After further rinsing, the samples were treated with 2% aqueous osmium tetroxide for 30 min, rinsed and en-bloc stained in 1% uranyl acetate for 2.5 hr and Walton's Lead for 30 min at 60°C. Samples were dehydrated in an ethanol series and impregnated with Durcupan ACM resin (Sigma) and mounted on Gatan 3View aluminium pins using conductive glue (CircuitWorks Conductive Epoxy). Samples

were gold coated and placed inside a Jeol field emission scanning electron microscope (JSM-7100F) equipped with a 3View 2XP system (Gatan). Section thickness was set at 45 nm and samples were imaged at 1.7kV under high vacuum using a 3048 × 2500 scan rate, to give a final pixel size of 31 nm.

### Laser cutting

Cell-cell interfaces within the roof plate of a Cdh2:(Cdh2-tFT) embryo were imaged and cut using a multi-photon microscope (Nikon A1R MP). Each cell boundary is a polygon with vertices linked by connecting cell interfaces. The mid-point on a connection between a pair of vertices was selected as a target for laser incision. Time-lapse, single z-level images of pre- and post-cut membranes were acquired at two frames per second. Recoil velocities were calculated from rate of change in the distance between the two vertices that had been linked by the connecting interface.

## Acknowledgements

We thank Claire Bromley, Clare Buckley, Jeremy Green, Rachel Moore, Christopher Rookyard, Andy Symonds and Vineetha Vijayakumar for helpful discussions and comments on the manuscript. We thank Darren Gilmour for the Cdh2:(Cdh2-tFT) line, and Masahiko Hibi for the Tg(atoh1a:EGFP) line. We are grateful to Gema Vizcay for her excellent support with 3View microscopy. This work was supported by UK BBSRC grant BB/I021507/1 (RW and JC) and a Wellcome Investigator Award (JC).

## Additional information

### Funding

| Funder | Grant reference number | Author |
|---|---|---|
| Biotechnology and Biological Sciences Research Council | BB/I021507/1 | Jonathan DW Clarke<br>Richard JT Wingate |
| Wellcome | 102895/Z/13/Z | Jonathan DW Clarke |

The funders had no role in study design, data collection and interpretation, or the decision to submit the work for publication.

### Author contributions

Florent Campo-Paysaa, Resources, Data curation, Formal analysis, Investigation, Methodology, Writing—original draft; Jonathan DW Clarke, Funding acquisition, Validation, Investigation, Visualization, Methodology, Writing—original draft, Project administration, Writing—review and editing; Richard JT Wingate, Conceptualization, Methodology, Writing—original draft, Project administration, Writing—review and editing

### Author ORCIDs

Richard JT Wingate (iD) http://orcid.org/0000-0002-1662-6097

### Ethics

Animal experimentation: This study was performed in strict accordance with the local ethics committee of King's College London and all animals handled according to the provisions of the Home Office UK Animals Scientific Procedures act 1986 (licence P70880F4C).

### Decision letter and Author response

Decision letter https://doi.org/10.7554/eLife.38485.020
Author response https://doi.org/10.7554/eLife.38485.021

## Additional files

### Supplementary files
• Transparent reporting form
DOI: https://doi.org/10.7554/eLife.38485.018

### Data availability
All data generated or analysed during this study are included in the manuscript and supporting files.

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
