## [Decision Letter]

Thank you for submitting your article "Generation of the squamous epithelial roof of the 4th ventricle" for consideration by *eLife*. Your article has been reviewed by three peer reviewers, and the evaluation has been overseen by Tanya Whitfield as Reviewing Editor and Marianne Bronner as the Senior Editor. The following individuals involved in review of your submission have agreed to reveal their identity: Raman M Das (Reviewer #1); Cecilia B Moens (Reviewer #2); Yevgenya Grinblat (Reviewer #3).

The reviewers have discussed the reviews with one another and the Reviewing Editor has drafted this decision to help you prepare a revised submission.

Summary:

The manuscript by Campo-Paysaa, Clarke and Wingate describes the generation of the squamous cell epithelium of the rhombencephalic roof plate in the developing zebrafish embryo. The work identifies a new cell type – the veil cell, which lies at the interface between the columnar progenitor cells and the squamous roof plate. This work proposes that these cells are the primary source for generation of squamous epithelia cells and that mechanical cues influence the orientation of veil cell divisions and also restrict the number of cells in the squamous roof plate.

Essential revisions:

All three reviewers are positive about the work, but all have queries for further clarification or suggestions for improvement to the manuscript. Many of these can be addressed by changes to the text and/or further analysis of existing data. As a minimum, full details of quantification are required (how many cells imaged, from how many embryos, and analysis of cell divisions from existing datasets).

The reviewers also make suggestions for further experimental work to support claims or speculation about the origin, fate and function of veil cells. After discussion, it was agreed that further experimental work is not essential prior to acceptance. However, if no further experimental support is provided, the authors must adjust the text accordingly throughout. In particular, they should tone down their strong claims about the long-term fate of the veil cells, their function as the stem cell pool for the entire roof plate, their role in maintaining the rhombic lip, and the claim that excess veil cells are eliminated by extrusion.

The full reviews are appended below for further information. Please address all the points that can be tackled without further experimental work. Suggestions from the reviewers requiring further experimental work can be completed at the authors' discretion.

*Reviewer #1:*

Summary

Overall, this is an elegant study based on some very nice observations. The experiments are well-designed and convincing for the most part. Conceptually, the idea of a transition cell-type that acts as the link between two different types of epithelia is interesting and is likely to be the case in several other tissue types. The concept of mechanical forces influencing division orientations is also timely and should prove to be of broad interest to readers beyond the neurodevelopment field.

Comments:

1) The authors state that the precise rostro-caudal limits of the ventricle are established by 36hpf, but there is no data or citations to support this. If this is known it should be cited or the data shown.

2) The ParD3 interfaces between the squamous epithelia and the apical surfaces of the columnar progenitor would be better presented with the addition of a 3D reconstruction of the image.

3) The drawings shown in Figure 5B are stated to be from a time-lapse movie. The relevant images from this movie should also be presented in the figure to back up this model of microtubule rearrangement.

4) Figure 6A maps the distribution of divisions in the roof plate. Is this a representative map of divisions from a single embryo? Were more embryos imaged? It would be useful to know the numbers of divisions observed and how consistent this is across different embryos.

5) The authors state that divisions within the squamous epithelium were extremely rare but again don't appear to provide cell / embryo numbers.

6) The authors propose that the direction of tension influences the orientation of veil cell divisions and that most asymmetrically fated divisions occur in the medio-lateral region. It would be useful to know if this continues to be the case if a neighbouring veil cell is lost, thus creating some extra space (perhaps by laser ablation?). Can the cells then switch to a symmetric mode of division to fill the space, or does the division orientation continue to be affected by the direction of tension?

*Reviewer #2:*

Campo-Paysaa et al. describe a novel cell type in the zebrafish hindbrain, where the pseudostratified columnar neuroepithelium meets the squamous roofplate epithelium. These elongated squamous "veil cells" act as a progenitor pool that populates the expanding roof plate either by direct transformation or by asymmetric cell division. Several interesting aspects of veil cell biology are described: their distinctive linear microtubule organization; their modes of cell division and the cortical tension conditions that may influence them; and their mode of elimination under crowding conditions. The authors show that the veil cells uniquely express gdf6a and suggest that this may be the Gdf source that sustains the underlying rhombic lip population.

The description of any new cell type is an event worth sitting up and taking notice of. For example, in *eLife* in 2017 Galanternik et al. described a novel perivascular cell type in zebrafish (PMC5423774). The current paper could be equally significant if more were known about the origins and eventual fates of the veil cells. However, without specific markers of the veil cells or the roof plate squamous epithelium, the authors are limited to studying these cells by live imaging for only the first ~12 hours of their existence. Consequently, some of the claims that are made based on this live imaging window are insufficiently supported by the data and should be strengthened.

Major criticisms:

1) The authors set out to answer the question "what is the origin of the squamous cells of the ventricular roof plate?" They describe a mesenchymal population already present at the dorsal midline at 17 hpf that appears to seed the roofplate, and, beginning at 22 hpf, a lateral population of veil cells that continue to contribute to the roofplate. It would be interesting to know the sources of both of these cells: did they delaminate at an earlier stage from the neuroepithelium? Are they a subset of neural crest cells which failed to migrate away? Retrospective timelapse analysis of sparsely mosaic embryos from earlier stages still should allow their origins to be determined.

2) In movie 4 the time points are too far apart to allow dots to be assigned to cells with certainty. For instance, the blue dot follows one nucleus from frames 1-10, then appears to switch to another nucleus, clearly neuroepithelial, which divides between frames 13 and 14. Then at frame 15 the blue dot appears to switch back to the original cell, which never divided, while the two daughters of the cell division reintegrate as neural progenitors. In the discussion, the authors state that some dorsal mesenchymal cells are able to integrate into the neural tube, although they do not indicate which data this is based on. Is it based on this blue cell? Such integration events should be more clearly demonstrated with closer timepoints and more sparsely labeled cells to support this important conclusion.

3) The specific expression of gdf6a in the veil cell population is beautifully demonstrated. Based on this and previous work showing that roofplate-derived Gdf (Gdf7) is sufficient for inducing rhombic lip identity, the authors conclude that "veil cells are a…signaling center that comprises the elusive organizer population for the rhombic lip". Loss-of-function, either by ablation of the veil cells or loss of gdf6a, is needed to support this strong conclusion.

4) The authors show by timelapse imaging that veil cells can both transform directly into roofplate squamous cells and that they can divide asymmetrically to give rise to roofplate cells. They also show that cell divisions are rare within the roofplate epithelium itself. The absence of cell divisions in the roofplate itself is notable given that this population will, over the next several days, give rise to a highly cellularized choroid plexus. While it is clear that the veil cells contribute to the roofplate, it is impossible to conclude, based on relatively short timelapses during the second day of development, that veil cell divisions are the source of all roofplate cells (Discussion, subsection “Veil cells”). Simple phospho-histone staining during the first five days of development could support the idea that the roofplate epithelial cells are post-mitotic, while more difficult lineage marking of individual veil cells could demonstrate the extent of their colonization of the roofplate. Without more extended analysis, the ultimate fate of veil cell progeny remains uncertain.

5) During the ~12 hour period being imaged starting at 22 hpf, both direct transformation events and asymmetric divisions events predominate in the lower rhombic lip while symmetric divisions predominate in the upper rhombic lip. This difference is attributed to higher tension along the mediolateral axis of the roofplate. However, this is not a sustainable situation, or else veil cells would gradually become depleted in the lower rhombic lip and accumulate in the upper rhombic lip. Continued timelapse imaging at later stages may reveal changes in the behaviors of veil cells at different positions along the roofplate boundary that would right this imbalance. More precise, quantitative data is required to support the authors' conclusions. For instance, the authors show only individual instances of division angle in symmetric and asymmetric veil cell divisions (Figure 6B and 6D). A quantitative analysis of the angles of all observed divisions should be provided to support the authors' assertions that cell division angle correlates with cell position and daughter cell fate. Additionally, in Figure 6G the authors combine all tension measurements across the entire tissue into one group. It is not safe to assume that tension is the same across the whole roofplate; the authors should bin the membrane retraction rates according to the location of the laser cut: upper rhombic lip region versus lower rhombic lip. Is the same mediolateral bias detected in each area?

6) Finally, using an ingenious experimental approach the authors show that reducing the size of the roofplate results in the extrusion of veil cells rather than a slowing of veil cell divisions due to reduced demand. To be able to conclude that the increase in extrusion is due to crowding (and not, for example, to the loss of trophic cues from r3/5) the authors should show that these events are prevented by transiently blocking cell division using the pharmacological method they developed for preventing neuroepithelial cell divisions. Furthermore, while it is clear from their dramatic fragmentation that the excess veil cells are eliminated, without 3-dimensional images it is impossible to say that this is really an extrusion event, and whether extrusion is apical or basal (which themselves are distinctly regulated processes).

*Reviewer #3:*

Campo-Paysaa et al. use zebrafish to full advantage, combining live imaging and genetic methods to address an important unanswered question: the mechanism of formation of the specialized epithelium at the roof of the hindbrain ventricle. Their approach proves fruitful, as it identifies a new epithelial stem/progenitor population that the authors term veil cells. Veil cells have unique properties and serve as a self-renewing progenitor pool for the squamous epithelium of the dorsal hindbrain. Overall, this is an elegant and convincing body of work that offers significant new insights into both our understanding epithelial growth in general, and of neural tube ventricle morphogenesis in particular. I recommend publication after the following minor problems are addressed:

The ability of veil cells to function as an organizer for the adjacent columnar neuroepithelium of the rhombic lip is overstated. For example, the sentence This population, which we name the "veil" cell for its distinctive morphology, also expresses gdf6a and is thus both the origin of roof plate development, and the cell type that confers organising signals to precursors the dorsal midline of the hindbrain neural tube" implies that organizing properties of veil cells have been experimentally tested, which is not the case. They were inferred from the fact that zebrafish veil cells uniquely express gdf6a, together with the demonstration of organizing properties of gdf7-expressing cells in chick embryos (an earlier work by the same group). To the best of my knowledge, equivalent organizing properties for gdf6a-positive cells have not been demonstrated in zebrafish. Until this is done, it must be made clear that veil cells are likely to have organizer properties, but that this remains be tested experimentally. This also applies to the first paragraph in the Discussion section.

---

## [Author Response]

Essential revisions:All three reviewers are positive about the work, but all have queries for further clarification or suggestions for improvement to the manuscript. Many of these can be addressed by changes to the text and/or further analysis of existing data. As a minimum, full details of quantification are required (how many cells imaged, from how many embryos, and analysis of cell divisions from existing datasets).The reviewers also make suggestions for further experimental work to support claims or speculation about the origin, fate and function of veil cells. After discussion, it was agreed that further experimental work is not essential prior to acceptance. However, if no further experimental support is provided, the authors must adjust the text accordingly throughout. In particular, they should tone down their strong claims about the long-term fate of the veil cells, their function as the stem cell pool for the entire roof plate, their role in maintaining the rhombic lip, and the claim that excess veil cells are eliminated by extrusion.The full reviews are appended below for further information. Please address all the points that can be tackled without further experimental work. Suggestions from the reviewers requiring further experimental work can be completed at the authors' discretion.Reviewer #1:[…] Comments:1) The authors state that the precise rostro-caudal limits of the ventricle are established by 36hpf, but there is no data or citations to support this. If this is known it should be cited or the data shown.

We have removed this sentence.

2) The ParD3 interfaces between the squamous epithelia and the apical surfaces of the columnar progenitor would be better presented with the addition of a 3D reconstruction of the image.

We tried several different reconstructions and none gave, in our view, a clearer view of the relationship of ParD3 to the epithelial surface than the pictures we present.

3) The drawings shown in Figure 5B are stated to be from a time-lapse movie. The relevant images from this movie should also be presented in the figure to back up this model of microtubule rearrangement.

Images of microtubule reorganisation were not a time-lapse sequence and we agree this could be misleading. We have therefore removed these images, drawings of microtubule organisation and the associated text.

4) Figure 6A maps the distribution of divisions in the roof plate. Is this a representative map of divisions from a single embryo? Were more embryos imaged? It would be useful to know the numbers of divisions observed and how consistent this is across different embryos.

This is a cumulative map of divisions from 125 embryos and this information has been added to the Figure legend on Figure 6.

5) The authors state that divisions within the squamous epithelium were extremely rare but again don't appear to provide cell / embryo numbers.

We observed 11 divisions in 124 movies (1100 hours).

Results section: “By contrast, we only observed 11 cell divisions within the squamous epithelium itself over the course of 124 time-lapse recordings (approximately 1100 hours), suggesting that such mitoses are extremely rare.

6) The authors propose that the direction of tension influences the orientation of veil cell divisions and that most asymmetrically fated divisions occur in the medio-lateral region. It would be useful to know if this continues to be the case if a neighbouring veil cell is lost, thus creating some extra space (perhaps by laser ablation?). Can the cells then switch to a symmetric mode of division to fill the space, or does the division orientation continue to be affected by the direction of tension?

This is an excellent question but answering it would be well beyond the scope of this study. Although we can create small laser cuts, discrete removal of cells by targeted ablation is not possible in our hands at least.

Reviewer #2:[…] The description of any new cell type is an event worth sitting up and taking notice of. For example, in eLife in 2017 Galanternik et al. described a novel perivascular cell type in zebrafish (PMC5423774). The current paper could be equally significant if more were known about the origins and eventual fates of the veil cells. However, without specific markers of the veil cells or the roof plate squamous epithelium, the authors are limited to studying these cells by live imaging for only the first ~12 hours of their existence. Consequently, some of the claims that are made based on this live imaging window are insufficiently supported by the data and should be strengthened.Major criticisms:1) The authors set out to answer the question "what is the origin of the squamous cells of the ventricular roof plate?" They describe a mesenchymal population already present at the dorsal midline at 17 hpf that appears to seed the roofplate, and, beginning at 22 hpf, a lateral population of veil cells that continue to contribute to the roofplate. It would be interesting to know the sources of both of these cells: did they delaminate at an earlier stage from the neuroepithelium? Are they a subset of neural crest cells which failed to migrate away? Retrospective timelapse analysis of sparsely mosaic embryos from earlier stages still should allow their origins to be determined.

This is indeed an excellent question. We have tried to answer this by looking at a Sox10 transgenic line which marks migrating neural crest cells. Because the reporter is expressed at low levels not only in roof plate but also neuroepithelial progenitors we really could not discern an answer to this question. Recording mosaic labelling in time-lapse movies in this early window proves difficult and the collection of a sufficient resource of films will be the goal of subsequent studies (along with a more efficient Sox10 marker). For the moment we cannot draw a conclusion.

2) In movie 4 the time points are too far apart to allow dots to be assigned to cells with certainty. For instance, the blue dot follows one nucleus from frames 1-10, then appears to switch to another nucleus, clearly neuroepithelial, which divides between frames 13 and 14. Then at frame 15 the blue dot appears to switch back to the original cell, which never divided, while the two daughters of the cell division reintegrate as neural progenitors. In the discussion, the authors state that some dorsal mesenchymal cells are able to integrate into the neural tube, although they do not indicate which data this is based on. Is it based on this blue cell?

We thank the referee for careful checking the reliability of our tracking and we agree it was difficult from the movie presented. We have now reduced the z-depth in the movie and highlighted only two cells that we believe can be unambiguously tracked. We have also improved the clarity of the images in the Figure to reflect these changes.

Results section: “Irregularly shaped midline cells undergo rostrocaudal movements which are, in rare examples, sufficiently pronounced to transverse rhombomeres boundaries, as if attracted towards regions where ventricle expansion is taking place”

Such integration events should be more clearly demonstrated with closer timepoints and more sparsely labeled cells to support this important conclusion.

See point 1. These early integration and delamination events will require a separate study with specific labelling strategies

3) The specific expression of gdf6a in the veil cell population is beautifully demonstrated. Based on this and previous work showing that roofplate-derived Gdf (Gdf7) is sufficient for inducing rhombic lip identity, the authors conclude that "veil cells are a…signaling center that comprises the elusive organizer population for the rhombic lip". Loss-of-function, either by ablation of the veil cells or loss of gdf6a, is needed to support this strong conclusion.

This conclusion has been down-played.

Results section: “This suggests that veil cells comprise a single cell wide signalling centre that organises the adjacent rhombic lip.”

4) The authors show by timelapse imaging that veil cells can both transform directly into roofplate squamous cells and that they can divide asymmetrically to give rise to roofplate cells. They also show that cell divisions are rare within the roofplate epithelium itself. The absence of cell divisions in the roofplate itself is notable given that this population will, over the next several days, give rise to a highly cellularized choroid plexus. While it is clear that the veil cells contribute to the roofplate, it is impossible to conclude, based on relatively short timelapses during the second day of development, that veil cell divisions are the source of all roofplate cells (Discussion, subsection “Veil cells”). Simple phospho-histone staining during the first five days of development could support the idea that the roofplate epithelial cells are post-mitotic.

The possibility that there might be a late wave of proliferation within the roof plate certainly a fascinating avenue to follow up on within a subsequent study. We have avoided extending this current study into later stages of development and choroid plexus development as this is not the focus of our study. In addition to proliferation at late stages (which we have yet to see), roof plate cells undergo significant modification of their cadherin rich boundaries. Considering proliferation without the associated events of choroid plexus formation would give a very partial picture of a complex patterning event.

While more difficult lineage marking of individual veil cells could demonstrate the extent of their colonization of the roofplate. Without more extended analysis, the ultimate fate of veil cell progeny remains uncertain.

As above, this examination of choroid plexus formation and its relation to veil cells is in our opinion a distinct question worthy of a separate, focused study. Genetic lineages suggest that veil cells may indeed be the ultimate origin of the choroid plexus, but mosaic labelling (with kaede) would be an excellent proof.

5) During the ~12 hour period being imaged starting at 22 hpf, both direct transformation events and asymmetric divisions events predominate in the lower rhombic lip while symmetric divisions predominate in the upper rhombic lip. This difference is attributed to higher tension along the mediolateral axis of the roofplate. However, this is not a sustainable situation, or else veil cells would gradually become depleted in the lower rhombic lip and accumulate in the upper rhombic lip.

Asymmetric divisions will not deplete the veil cell population and we show that some symmetric divisions do occur in the lower rhombic lip: Veil cells shouldn’t be depleted in the short term in this location. Direct transformation is a slow and relatively infrequent process and so cell loss from the rhombic lip is unlikely to be a major confounding factor in balancing growth. Of course, expansion of the roof plate stops at some point. We have not investigated this however, it is intriguing to speculate that this could be related to reduced tension as the underlying ventricle ceases to expand.

Continued timelapse imaging at later stages may reveal changes in the behaviors of veil cells at different positions along the roofplate boundary that would right this imbalance.

We agree that this is a good suggestion but consider it outside the scope of our current study. We also feel that these questions will be best tested within an in silico model which can be manipulated and then be compared against in vivo data.

More precise, quantitative data is required to support the authors' conclusions. For instance, the authors show only individual instances of division angle in symmetric and asymmetric veil cell divisions (Figure 6B and 6D). A quantitative analysis of the angles of all observed divisions should be provided to support the authors' assertions that cell division angle correlates with cell position and daughter cell fate.

We agree and have softened this issue in the relevant text. We have removed statements about cleavage angle and fate. We still believe it possible that daughter fate is related to the position of the daughters soon after mitosis and their relative position could be determined by local tension. This section “Veil cells can also generate squamous roof plate cells via asymmetric divisions and self-renew via symmetric divisions” is substantially rewritten.

Additionally, in Figure 6G the authors combine all tension measurements across the entire tissue into one group. It is not safe to assume that tension is the same across the whole roofplate; the authors should bin the membrane retraction rates according to the location of the laser cut: upper rhombic lip region versus lower rhombic lip. Is the same mediolateral bias detected in each area?

This data has been replotted to include a distinction between upper and lower rhombic lip in Figure 6G and H and associated legends. There is no significant difference between the median recoil velocity in upper and lower rhombic lip.

Legend Figure 6:: Quantification of initial recoil velocity after laser cut as a function of the angle to the anteroposterior axis shows a significantly higher velocity following parallel versus perpendicular cuts (Two-tailed Mann-Whitney, p=0.0113). Recoil in the region proximal to the upper rhombic lip (orange) is similar to that adjacent to the lower rhombic lip (blue). H. Map of positions and angles of laser cuts.

6) Finally, using an ingenious experimental approach the authors show that reducing the size of the roofplate results in the extrusion of veil cells rather than a slowing of veil cell divisions due to reduced demand. To be able to conclude that the increase in extrusion is due to crowding (and not, for example, to the loss of trophic cues from r3/5) the authors should show that these events are prevented by transiently blocking cell division using the pharmacological method they developed for preventing neuroepithelial cell divisions. Furthermore, while it is clear from their dramatic fragmentation that the excess veil cells are eliminated, without 3-dimensional images it is impossible to say that this is really an extrusion event, and whether extrusion is apical or basal (which themselves are distinctly regulated processes).

This is an excellent suggestion but it would be complicated by the fact that this will also alter the growth of the underlying neuroepithelium. It would be thus also be tricky to interpret. We feel that we offer an appropriately cautious interpretation of our current data in suggesting that space restriction may regulate squamous cell number. We do not claim that we have definitively demonstrated this to be the case.

Reviewer #3:Campo-Paysaa et al. use zebrafish to full advantage, combining live imaging and genetic methods to address an important unanswered question: the mechanism of formation of the specialized epithelium at the roof of the hindbrain ventricle. Their approach proves fruitful, as it identifies a new epithelial stem/progenitor population that the authors term veil cells. Veil cells have unique properties and serve as a self-renewing progenitor pool for the squamous epithelium of the dorsal hindbrain. Overall, this is an elegant and convincing body of work that offers significant new insights into both our understanding epithelial growth in general, and of neural tube ventricle morphogenesis in particular. I recommend publication after the following minor problems are addressed:The ability of veil cells to function as an organizer for the adjacent columnar neuroepithelium of the rhombic lip is overstated. For example, the sentence This population, which we name the "veil" cell for its distinctive morphology, also expresses gdf6a and is thus both the origin of roof plate development, and the cell type that confers organising signals to precursors the dorsal midline of the hindbrain neural tube" implies that organizing properties of veil cells have been experimentally tested, which is not the case. They were inferred from the fact that zebrafish veil cells uniquely express gdf6a, together with the demonstration of organizing properties of gdf7-expressing cells in chick embryos (an earlier work by the same group). To the best of my knowledge, equivalent organizing properties for gdf6a-positive cells have not been demonstrated in zebrafish. Until this is done, it must be made clear that veil cells are likely to have organizer properties, but that this remains be tested experimentally. This also applies to the first paragraph in the Discussion section.

This conclusion has been down-played.

Results section: “This suggests that veil cells comprise a single cell wide signalling centre that organises the adjacent rhombic lip.”

Discussion section: “On the assumption of a conservation of function across species, *Gdf6a*-positive veil cells may also act as a classic boundary organizer.”